# Development of a risk prediction model for sepsis-related delirium based on multiple machine learning approaches and an online calculator

Lang Gao[1], Guang Dong Wang[2], Xing Yi Yang[3], Shi Jun Tong[1], Xu Jie Wang[4], Yun Ruo Chen[1], Jin Ying Bai[1], Ya Xin Zhang [5]*

1 Department of Critical Care Medicine, Clinical Medical College of Qinghai University, Xi ning, China, 2 Department of Respiratory and Critical Care Medicine, First Affiliated Hospital of Xi'an Jiaotong University, Xi'an, Shanxi, China, 3 Department of Gastroenterology Disease, XianJu People's Hospital, Zhejiang Southeast Campus of Zhejiang Provincial People's Hospital, Affiliated Xianju's Hospital, Hang zhou Medical College, Xianju, Zhejiang, China, 4 Department of Emergency Medicine, Clinical Medical College of Qinghai University, Xi ning, China, 5 Department of Neurology, Xia men Humanity Hospital, Fujian Medical University, Xia men, Fujian, China

* 1738048108@qq.com

## Abstract

### Background

Sepsis-associated delirium (SAD) occurs due to disruptions in neurotransmission linked to inflammatory responses from infections. It poses significant challenges in clinical management and is associated with poor outcomes. Survivors often experience long-term cognitive and behavioral issues that impact their quality of life and place a burden on their families. This study aimed to develop and validate an interpretable machine learning model for early prediction of SAD in critically ill patients. Additionally, we constructed an online risk calculator to facilitate real-time clinical assessment.

### Methods

This study is a retrospective analysis utilizing data from 16,120 patients in the Medical Information Mart for Intensive Care IV database. To manage imbalanced data, we applied the Synthetic Minority Over-sampling Technique (SMOTE) method. Feature selection was conducted using Multivariate Logistic Regression, LASSO regression, and the Boruta algorithm. We developed predictive models using eight machine learning algorithms and selected the best one for validation. The SHapley Additive exPlanations (SHAP) method was used for visualization and interpretation, enhancing the clinical understanding of the model, alongside the creation of an online web calculator.

**Data availability statement:** All relevant data are within the manuscript and its Supporting information files.

**Funding:** The author(s) received no specific funding for this work.

**Competing interests:** The authors have declared that no competing interests exist.

**Abbreviations:** MIMIC-IV, Medical Information Mart for Intensive Care IV; SAD, Sepsis-associated delirium; MLR, Multivariate Logistic Regression; LASSO, Least Absolute Shrinkage and Selection Operator Regression; SHAP, The SHapley Additive exPlanations; SOFA, Sequential Organ Failure Assessment; GCS, Glasgow Coma Scale; SAPSII, Simplified Acute Physiology Score II; RDW, Red cell distribution width; AKI, Acute kidney injury; MV, Mechanical ventilation; CRRT, Continuous renal replacement therapy; CAM-ICU, Confusion Assessment Method for the Intensive Care Unit; SMOTE, Synthetic Minority Over-sampling Technique; LR, Logistic Regression; SVM, Support Vector Machine; GBM, Gradient Boosting Machine; RF, Random Forest; XGBoost, Extreme Gradient Boosting; LightGBM, Adaptive Boosting, AdaBoost, Light Gradient Boosting Machine; DCA, Decision Curve Analysis.

## Results

We combined three feature selection methods to identify 17 key features for our machine learning prediction model. The Gradient Boosting Machine (GBM) model demonstrated excellent calibration and strong predictive accuracy in the validation cohort. The SHAP feature importance ranking revealed five critical risk factors for predicting outcomes: Glasgow Coma Scale (GCS), ICU stay duration, chloride, sodium, and Sequential Organ Failure Assessment (SOFA). Based on this optimal model, we successfully developed an online web calculator.

## Conclusion

We developed and validated a machine learning model capable of accurately predicting SAD with high clinical applicability. The integration of interpretable machine learning and an online calculator offers a practical tool to support early identification and timely management of SAD in critically ill patients.

## 1. Introduction

Sepsis is a host's uncontrolled systemic inflammatory response syndrome triggered by infection, where a large number of inflammatory factors lead to multi-organ dysfunction, ultimately resulting in a crisis of multiple organ failure that can threaten the patient's life [1]. Although supportive bundled care for sepsis has been helpful in reducing mortality rates among sepsis patients, the overall prognosis remains poor. According to the Centers for Disease Control and Prevention, the annual incidence of sepsis in the United States is 300–1,000 cases per 100,000 people. Global burden of disease studies show that the global annual incidence of sepsis reaches up to 31 million cases, making it a major public health threat [2–4].

Delirium is an acute brain dysfunction characterized by fluctuations in attention and impaired cognitive function, with diverse symptoms that may include psychomotor agitation and altered consciousness. It is commonly seen in critically ill patients admitted to the intensive care unit (ICU) [5]. Sepsis is one of the major risk factors for delirium, as the systemic inflammatory response syndrome associated with sepsis can disrupt the balance of central nervous system function, leading to delirium in patients. Delirium in sepsis patients is referred to as sepsis-associated delirium (SAD); however, the mechanisms by which sepsis affects the central nervous system remain unclear and may involve brain inflammation, cerebral perfusion, blood-brain barrier disruption, and neurotransmitter disturbances. Managing SAD in the ICU has historically been challenging, with poor prognoses for SAD patients. Survivors often experience long-term and severe cognitive impairments and behavioral abnormalities, significantly reducing their quality of life and placing a heavy burden on families [6,7]. Therefore, early identification and prevention of potential SAD patients in clinical practice are of utmost importance. Consequently, we developed a predictive model for SAD and constructed an online calculator to provide clinicians with an

important tool for early identification of high-risk populations, optimizing individual intervention measures through various simple clinical indicators to improve clinical outcomes and reduce the length of hospital stay.

The rapid development of artificial intelligence and machine learning algorithms has significantly accelerated the innovation of predictive models for medical diagnosis and prognosis assessment of various diseases [8,9]. Compared to traditional regression analysis, machine learning is widely used to handle clinical data, extracting features related to clinical outcomes from large datasets and identifying independent predictive factors associated with clinical results, thereby better addressing clinical decision-making issues [10,11]. In the ICU, assessing a patient's risk of developing delirium heavily relies on subjective judgment by healthcare professionals, necessitating the development of a more straightforward, objective, and accurate tool to evaluate the risk of delirium. The aim of this study is to establish an interpretable machine learning-based online predictive tool for SAD risk to assist clinicians in accurately assessing the risk of delirium early, allowing timely adjustments to treatment plans to reduce the incidence of delirium and make more informed clinical decisions.

## 2. Materials and methods

### 2.1 Study design

The data for this study were sourced from the Medical Information Mart for Intensive Care III (MIMIC-III, version 1.4) and MIMIC-IV (version 3.1) databases, developed by the Laboratory for Computational Physiology at the Massachusetts Institute of Technology. These publicly available, de-identified databases comprise one of the largest and most comprehensive collections of critical care data worldwide. MIMIC-III includes detailed clinical records from approximately 60,000 ICU admissions at the Beth Israel Deaconess Medical Center between 2001 and 2012. MIMIC-IV expands on this scope, containing data from nearly 90,000 patients admitted between 2008 and 2022. Both databases provide a wide array of structured and unstructured clinical information, including demographic characteristics, diagnoses, procedures, medication use, laboratory test results, nursing documentation, and outcome data [12,13]. To utilize the database, we completed the web-based course provided by the National Institutes of Health (NIH), fulfilling the CITI program training requirements and obtaining research ethics certification (certification number: 66380198). Our study complies with the Declaration of Helsinki and international medical ethics standards, and all patient data were fully anonymized to waive informed consent requirements and ethical review.

### 2.2 Study population and outcome

We extracted each patient's hospitalization information, including demographics, vital signs, laboratory test indicators, and treatment information, from three databases using Structured Query Language (SQL). Patients included in the study met the following criteria: (1) confirmed or suspected infection within 24 hours of ICU admission, and according to the Sepsis-3.0 diagnostic criteria, a Sequential Organ Failure Assessment (SOFA) score of ≥2. (2) Patients were aged 18–100 years and had laboratory test records within 24 hours of ICU admission. (3) first admission to the ICU. (4) Patients assessed using the Confusion Assessment Method for the Intensive Care Unit (CAM-ICU). CAM-ICU is an effective screening tool for identifying delirium in the intensive care unit, and septic patients with positive assessments are defined as having SAD. The exclusion criteria were as follows: (1) patients with dementia or schizophrenia; (2) missing variables greater than 30%; (3) ICU stay of less than 24 hours; (4) outliers in vital signs. (5) develop delirium before admission to the ICU. The data collection process for the training and testing sets is illustrated in Fig 1.

### 2.3 Inclusion of variables

This retrospective study included 46 clinical variables extracted from the MIMIC-III and MIMIC-IV databases. The dataset comprised demographic information (age, sex, and weight), baseline clinical characteristics, and comorbidities. Vital

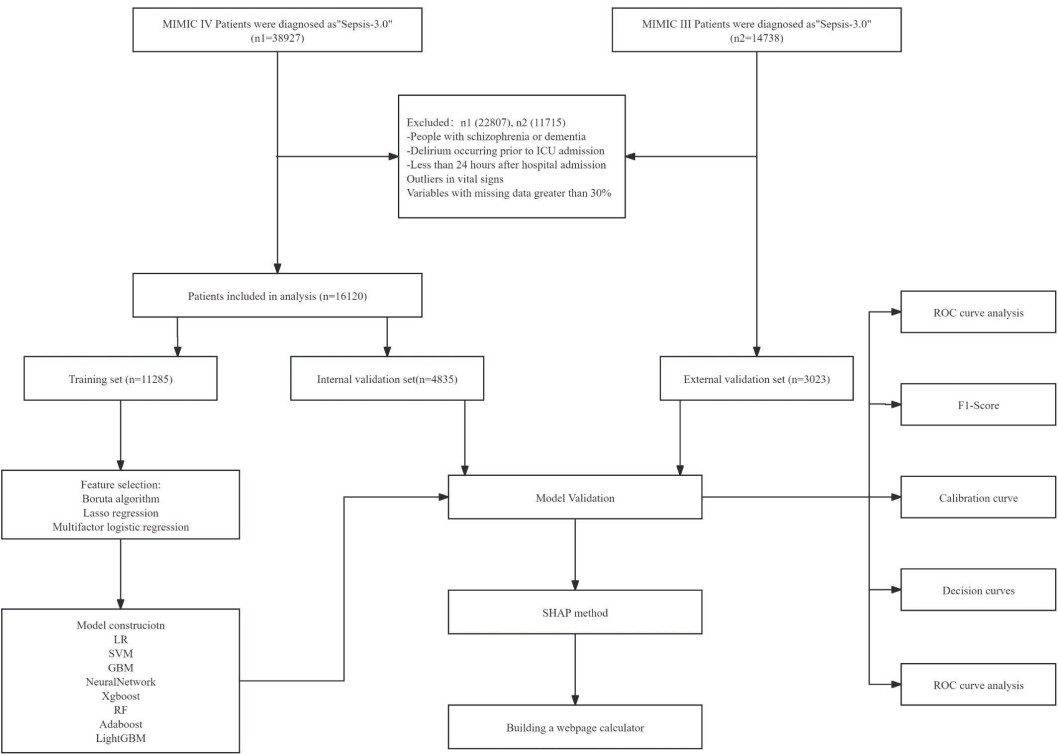

**Fig 1. The entire research flowchart.**

signs recorded within the first 24 hours of ICU admission included heart rate (HR), systolic blood pressure (SBP), diastolic blood pressure (DBP), mean arterial pressure (MAP), respiratory rate (RR), oxygen saturation (SpO$_2$), body temperature, Sequential Organ Failure Assessment (SOFA) score, Simplified Acute Physiology Score II (SAPS II), and Glasgow Coma Scale (GCS) score. Laboratory parameters measured within 24 hours of ICU admission encompassed hematological indices, red blood cell (RBC) count, white blood cell (WBC) count, platelet count, hemoglobin, hematocrit, red cell distribution width (RDW), as well as coagulation profiles (international normalized ratio, prothrombin time, activated partial thromboplastin time), renal function markers (creatinine, blood urea nitrogen), and metabolic indicators (anion gap, pH, bicarbonate, calcium, magnesium, chloride, potassium, sodium, lactate, and glucose). Recorded comorbidities included hypertension, acute kidney injury (AKI), type 2 diabetes mellitus, and heart failure. Iatrogenic and environmental factors comprised major therapeutic interventions during ICU stay, such as mechanical ventilation (MV), continuous renal replacement therapy (CRRT), and administration of medications including midazolam and vasopressin (VP). All variables had less than 20% missing data. Multiple imputation techniques were applied to address missingness, ensuring dataset completeness and minimizing potential bias.

## 2.4 The selection of features and the processing of data

In this study, This approach alleviates the bias introduced by sample imbalance while avoiding the risks associated with excessive oversampling. The training set was used for model development, while the validation set was employed to assess the model's performance. To determine potential predictive factors in the training set, we utilized three independent methods to filter baseline variables: Multiple Linear Regression (MLR), Least Absolute Shrinkage and Selection Operator (LASSO), and the Boruta algorithm [14]. MLR is a variable selection method based on covariate adjustment that controls

for confounding factors. By utilizing statistical significance (P < 0.05), MLR identifies variables independently associated with outcomes while retaining predictive factors that remain interpretable in the presence of multivariable co-occurrence, providing adjusted odds ratios (OR) and their confidence intervals. This process results in a validated variable set with both statistical and clinical significance, offering robust support for clinical decision-making [15]. LASSO regression, when handling high-dimensional data, applies penalties to feature coefficients, automatically selecting variables that have a practical impact on the prediction target. When multiple features are correlated, LASSO regression retains the most representative feature while eliminating multicollinearity interference [16,17]. To optimize model effectiveness, we employed 20-fold cross-validation to analyze baseline high-dimensional data and selected variables, which mitigated the risk of overfitting and ensured result stability. The Boruta algorithm is an ensemble feature selection framework based on random forests, where the core idea is to conduct significance testing on original variables by constructing "shadow features" [18]. The Boruta algorithm is capable of capturing complex nonlinear relationships, effectively reducing the risk of overfitting while enhancing model performance, thus providing a more reliable feature space for subsequent model construction [19]. To ensure maximum robustness of the selected features, mitigate overfitting risks, enhance clinical interpretability, and meet the requirements for both feature quantity and practicality in subsequent online calculators, this study ultimately selected the intersecting features identified by Boruta, LASSO, and MLR as the modeling feature set.

## 2.5 Model development and evaluation

We employed a random sampling method to allocate the patients included in the study into the training set and validation set in a ratio of 7:3. The data distribution in the training set was as follows: 2,355 SAD cases and 8,930 Non-SAD cases. Given that class imbalance may introduce bias into the machine learning prediction model, we conducted a supplementary analysis comparing the use of oversampling, undersampling, and the Synthetic Minority Over-sampling Technique (SMOTE) to balance the training set. Critically, SMOTE was applied exclusively to the training set. After balancing using the SMOTE algorithm, the final training set comprised 7,065 SAD cases and 7,065 Non-SAD cases. This approach effectively mitigates model bias induced by class imbalance while avoiding the risk associated with oversampling.

We utilized eight machine learning methods to construct models in the training set: Logistic Regression (LR), Support Vector Machine (SVM), Gradient Boosting Machine (GBM), Neural Network, Random Forest (RF), Extreme Gradient Boosting (XGBoost), Adaptive Boosting (AdaBoost), and Light Gradient Boosting Machine (LightGBM). In this study, we employed the "caret" package to perform model selection using 10-fold cross-validation, while incorporating a grid parameter optimization process during the model training phase, as detailed in S1 Table [20]. Subsequently, we tested the models' performance using both internal and external validation sets, evaluating different models through the Area Under the Receiver Operating Characteristic Curve (AUC), Decision Curve Analysis (DCA), and calibration curves. Additionally, we calculated the Accuracy, Sensitivity, and F1-score for each model to further assess performance. Based on the evaluation results, we selected the best-performing model for SHAP significance analysis and generated SHAP summary plots to assess feature importance. We then conducted SHAP dependence plots to analyze the mechanisms by which features influence the prediction outcomes, and finally quantified the contribution weights of each feature within individual samples using SHAP analysis. For ease of use by clinicians, we developed a user-friendly web calculator.

## 2.6 Statistical analysis

In this study, all statistical analyses were performed using R software (version 4.4.2) and Python software (version 3.10.6). To compare baseline characteristics between the two groups in the MIMIC-IV dataset, continuous variables conforming to a normal distribution were expressed as Mean ± Standard Deviation (x ± s), while non-normally distributed continuous variables were denoted as Median (Interquartile Range) [M (IQR)]. Continuous variables were analyzed using independent samples t-test or Wilcoxon rank-sum test. Categorical variables were expressed as numbers (percentage) [n (%)], and the $\chi^2$ test or Fisher's exact test was employed for categorical variables, with a two-sided P < 0.05 indicating statistical

significance. To address the issue of sample imbalance in the training set, we utilized the SMOTE algorithm based on the "DMwR" package in R. For model development, the dataset was randomly divided into a training set (70%) and an internal validation set (30%). Variable selection was performed using MLR, LASSO regression, and the Boruta algorithm, incorporating the shared variables from these three algorithms into the machine learning model. The "pROC" and "ggplot2" packages were employed to plot the ROC curve analysis and AUC values for the internal validation set, identifying the best predictive model. Finally, SHAP analysis was utilized to interpret the optimal model.

## 3. Results

### 3.1 Baseline characteristics

We collected data from 16,120 sepsis patients in the MIMIC-IV database, with detailed screening processes illustrated in Fig 1. Among the included sepsis patients, 3,364 (26.4%) were diagnosed with delirium. Table 1 summarizes the characteristics of SAD and Non-SAD patients in the MIMIC-IV database, including demographics, treatment information, and laboratory test results. Overall, the SAD group had higher values for Age, ICU Day, SOFA, SAPS II, Creatinine, Blood Urea Nitrogen, Anion Gap, Glucose, and Platelet counts compared to the Non-SAD group, with statistical significance (P < 0.05). The median age of the study population was 67 years (57, 77), with a gender distribution of 9,532 (59%) males. We divided the total population into a training set (70%) and an internal validation set (30%), utilizing the training set for model development. To explore the correlations between variables, we plotted a correlation bar chart for each variable (Fig 2A) and a heatmap of the correlations between variables (Fig 2B).

### 3.2 Variable selection

The 16,120 patients collected from the MIMIC-IV database were randomly divided into a training set (70%) and an internal validation set (30%). To exclude irrelevant variables, we performed an initial screening of variables using MLR, LASSO regression, and the Boruta algorithm. As shown in Table 2, ULR analysis indicated statistical relationships between AKI, Type 2 Diabetes Mellitus, Heart Failure, MV, CRRT, Midazolam, VP, Age, Weight, ICU Day, SOFA, SAPS II, GCS, SBP, DBP, MAP, Temperature, Platelet, RDW, MCHC, MCV, Creatinine, Blood Urea Nitrogen, Anion Gap, Calcium, Magnesium, Chloride, Sodium, Lactate, and Glucose with SAD patients (P < 0.05). MLR analysis of these results suggested that these factors have significant statistical associations with SAD patients, indicating they may be independent risk factors for patient mortality. The best LASSO regression lambda.1se value was confirmed to be 0.0035 through 20-fold cross-validation, resulting in the identification of 28 variables with significant predictive power (Fig 3A and 3B). Additionally, we utilized the Boruta algorithm for a more in-depth analysis to clarify key variables. This algorithm effectively distinguishes between strongly correlated and weakly correlated variables, significantly improving predictive accuracy. In the analysis results presentation, the green boxes indicate shadow features automatically generated by the algorithm, which were excluded from the final analysis to focus on the most influential variables. Ultimately, we identified 40 impactful variables (Fig 3C and 3D). Finally, we generated a Venn diagram using the "ggvenn" package in R, revealing 17 features shared among the three algorithms (Fig 3E). Based on these features, we established eight machine learning predictive models.

### 3.3 Model performance on test and external validation datasets

Upon identifying 17 clinical features, we employed eight distinct machine learning algorithms to construct predictive models for SAD. Following the application of the SMOTE function to balance the training set, the models developed on internal and external validation sets—specifically LR, SVM, GBM, Neural Network, RF, XGBoost, AdaBoost, and LightGBM—achieved AUC values of 0.67, 0.68, 0.73, 0.69, 0.71, 0.68, 0.69, and 0.70, respectively (Fig 4A). Corresponding AUC values for the external validation set were 0.61, 0.61, 0.70, 0.72, 0.70, 0.65, 0.66, and 0.67 (Fig 4B). Detailed performance metrics, including Accuracy, Sensitivity, Specificity, Precision, and F1-Score for both internal and external validation sets,

**Table 1. Baseline characteristics of SAD and Non-SAD patients.**

| Variables | Total (n = 16120) | NSAD (n = 12756) | SAD (n = 3364) | p |
|---|---|---|---|---|
| **Demographics** | | | | |
| Age(years) | 67 (57, 77) | 67 (57, 77) | 68 (57, 78) | 0.004 |
| Gender(%) | | | | 0.645 |
| Female | 6588 (41) | 5201 (41) | 1387 (41) | |
| Male | 9532 (59) | 7555 (59) | 1977 (59) | |
| **Comorbidity** | | | | |
| Hypertension(%) | | | | 0.003 |
| No | 9722 (60) | 7617 (60) | 2105 (63) | |
| Yes | 6398 (40) | 5139 (40) | 1259 (37) | |
| AKI(%) | | | | < 0.001 |
| No | 9187 (57) | 7499 (59) | 1688 (50) | |
| Yes | 6933 (43) | 5257 (41) | 1676 (50) | |
| T2DM(%) | | | | 0.197 |
| No | 11437 (71) | 9081 (71) | 2356 (70) | |
| Yes | 4683 (29) | 3675 (29) | 1008 (30) | |
| HF(%) | | | | < 0.001 |
| No | 11451 (71) | 9140 (72) | 2311 (69) | |
| Yes | 4669 (29) | 3616 (28) | 1053 (31) | |
| **ICU interventions** | | | | |
| MV(%) | | | | < 0.001 |
| No | 2102 (13) | 1726 (14) | 376 (11) | |
| Yes | 14018 (87) | 11030 (86) | 2988 (89) | |
| CRRT(%) | | | | 0.002 |
| No | 14681 (91) | 11663 (91) | 3018 (90) | |
| Yes | 1439 (9) | 1093 (9) | 346 (10) | |
| Midazolam(%) | | | | < 0.001 |
| No | 12249 (76) | 9826 (77) | 2423 (72) | |
| Yes | 3871 (24) | 2930 (23) | 941 (28) | |
| VP(%) | | | | 0.026 |
| No | 5549 (34) | 4336 (34) | 1213 (36) | |
| Yes | 10571 (66) | 8420 (66) | 2151 (64) | |
| **Scoring system** | | | | |
| SOFA(score) | 6 (4, 8) | 5 (3, 8) | 6 (4, 9) | < 0.001 |
| SAPSII (score) | 38 (30, 49) | 38 (29, 48) | 41 (33, 51) | < 0.001 |
| GCS (score) | 15 (13, 15) | 15 (14, 15) | 14 (11, 15) | < 0.001 |
| **Vital signs** | | | | |
| HR(minute) | 85.24 (75.32, 97.45) | 85.04 (75.28, 97.13) | 85.91 (75.54, 98.62) | 0.006 |
| SBP(mmHg) | 111.18 (102.27, 123.36) | 110.88 (102.1, 122.92) | 112.67 (102.82, 125.47) | < 0.001 |
| DBP(mmHg) | 62.67 (56.14, 70) | 62.43 (56, 69.68) | 63.67 (57, 71) | < 0.001 |
| MAP(mmHg) | 75 (68.67, 82.77) | 74.75 (68.41, 82.35) | 76 (69.7, 84.34) | < 0.001 |
| RR(minute) | 19.15 (16.85, 22.2) | 19.04 (16.77, 22.13) | 19.45 (17.07, 22.46) | < 0.001 |
| SPO2(%) | 97.17 (95.68, 98.5) | 97.17 (95.68, 98.5) | 97.14 (95.68, 98.52) | 0.645 |
| Temperature(°C) | 36.87 (36.66, 37.17) | 36.86 (36.65, 37.15) | 36.93 (36.68, 37.28) | < 0.001 |
| ICU day(days) | 3.44 (1.93, 7.06) | 3.14 (1.82, 6.39) | 4.9 (2.59, 9.71) | < 0.001 |

*(Continued)*

**Table 1.** (Continued)

| Variables | Total (n = 16120) | NSAD (n = 12756) | SAD (n = 3364) | p |
|---|---|---|---|---|
| **Laboratory Results** | | | | |
| RBC($10^{12}$/L) | 3.43 (2.97, 3.95) | 3.43 (2.98, 3.94) | 3.41 (2.92, 3.97) | 0.089 |
| WBC($10^9$/L) | 12 (8.6, 16.2) | 12.05 (8.68, 16.2) | 11.8 (8.35, 16.3) | 0.18 |
| Platelet($10^9$/L) | 171 (121.24, 237.37) | 169 (121, 235.08) | 178 (122.19, 246) | < 0.001 |
| Hemoglobin(g/dL) | 10.23 (8.81, 11.75) | 10.25 (8.87, 11.73) | 10.15 (8.67, 11.8) | 0.093 |
| RDW(%) | 14.66 (13.5, 16.47) | 14.6 (13.45, 16.3) | 14.97 (13.7, 17) | < 0.001 |
| HCT(%) | 31.3 (27.3, 35.78) | 31.3 (27.34, 35.7) | 31.3 (27.1, 36.14) | 0.573 |
| MCH(pg picograms) | 30.15 (28.58, 31.58) | 30.15 (28.6, 31.53) | 30.1 (28.5, 31.77) | 0.537 |
| MCHC(g/dL) | 32.75 (31.67, 33.75) | 32.8 (31.73, 33.8) | 32.5 (31.43, 33.53) | < 0.001 |
| MCV(fL femtoliters) | 91.67 (87.67, 96) | 91.5 (87.5, 95.67) | 92.33 (88, 97) | < 0.001 |
| INR(ratio) | 1.3 (1.2, 1.58) | 1.3 (1.2, 1.57) | 1.3 (1.15, 1.6) | 0.842 |
| PT(seconds) | 14.3 (12.7, 17.1) | 14.3 (12.75, 17) | 14.3 (12.5, 17.5) | 0.763 |
| APTT(seconds) | 31.68 (27.9, 39.5) | 31.75 (28, 39.5) | 31.38 (27.62, 39.39) | 0.024 |
| Creatinine(mg/dL) | 1.05 (0.77, 1.7) | 1.05 (0.75, 1.67) | 1.1 (0.77, 1.97) | < 0.001 |
| BUN(mg/dL) | 21 (14, 35.67) | 20.5 (14, 34.5) | 23.5 (14.5, 40) | < 0.001 |
| Anion Gap(mEq/L) | 14 (12, 16.67) | 14 (11.67, 16.5) | 14.25 (12, 17) | < 0.001 |
| PH | 7.37 (7.33, 7.41) | 7.37 (7.33, 7.41) | 7.37 (7.32, 7.41) | 0.066 |
| Bicarbonate(mEq/L) | 22.25 (19.67, 24.5) | 22.33 (19.67, 24.5) | 22 (19.33, 25) | 0.213 |
| Calcium(mg/dL) | 8.23 (7.8, 8.7) | 8.2 (7.8, 8.7) | 8.3 (7.85, 8.8) | < 0.001 |
| Magnesium(mg/dL) | 2.03 (1.85, 2.27) | 2.04 (1.85, 2.28) | 2.02 (1.85, 2.25) | 0.005 |
| Chloride(mEq/L) | 104.4 (100, 108) | 104.5 (100.5, 108) | 104 (99.33, 107.5) | < 0.001 |
| Potassium(mEq/L) | 4.18 (3.85, 4.56) | 4.2 (3.85, 4.55) | 4.13 (3.8, 4.6) | 0.008 |
| Sodium(mEq/L) | 138.33 (135.5, 141) | 138 (135.5, 140.5) | 139 (135.67, 141.87) | < 0.001 |
| Lactate(mmol/L) | 1.83 (1.33, 2.6) | 1.85 (1.35, 2.61) | 1.8 (1.3, 2.6) | < 0.001 |
| Glucose(mg/dL) | 129 (108.33, 162.5) | 128.54 (108.33, 161.33) | 131 (108, 166.5) | 0.02 |

Abbreviations: SOFA, Sequential Organ Failure Assessment; SAPSII, Simplified Acute Physiologic Score II; GCS, Glasgow Coma Scale; SBP, Systolic blood pressure; DBP, Diastolic blood pressure; MAP, Mean arterial pressure; RDW, Red cell distribution width; MCHC, Mean corpuscular hemoglobin concentration; MCV, Mean corpuscular volume; BUN, Blood urea nitrogen; AKI, Acute kidney injury; T2DM, Type 2 diabetes mellitus; HF, Heart failure; MV, Mechanical ventilation; RBC, Red blood cell; WBC, White blood cell; HCT, Hematocrit; MCH, Mean corpuscular hemoglobin; INR, International normalized ratio; PT, Prothrombin time; APTT, Activated partial thromboplastin time.

are provided in Table 3. Due to the substantial performance gap observed between the training and validation sets for the RF model, raising concerns about potential overfitting, the GBM model was ultimately selected as optimal. To further evaluate the GBM model's performance, we generated DCA curves for both the internal and external validation sets (Fig 4C–4F). The GBM model's calibration curve and DCA demonstrated robust predictive performance across the majority of threshold probability ranges and indicated significant clinical net benefit. Based on the comprehensive assessment of the above performance metrics, the GBM model was confirmed as the most suitable predictive tool for this dataset and was selected for subsequent analyses. A confusion matrix illustrating the GBM model's classification performance for patients in the internal validation set is presented (Fig 4G). Finally, the feature importance plot for the GBM model is also shown (Fig 4H).

### 3.4 Interpretability analysis

To further explore the clinical application of the GBM model, we utilized the SHAP algorithm to quantify the contribution of each feature within the model. The feature importance bee swarm plot we created illustrates the mechanisms by which

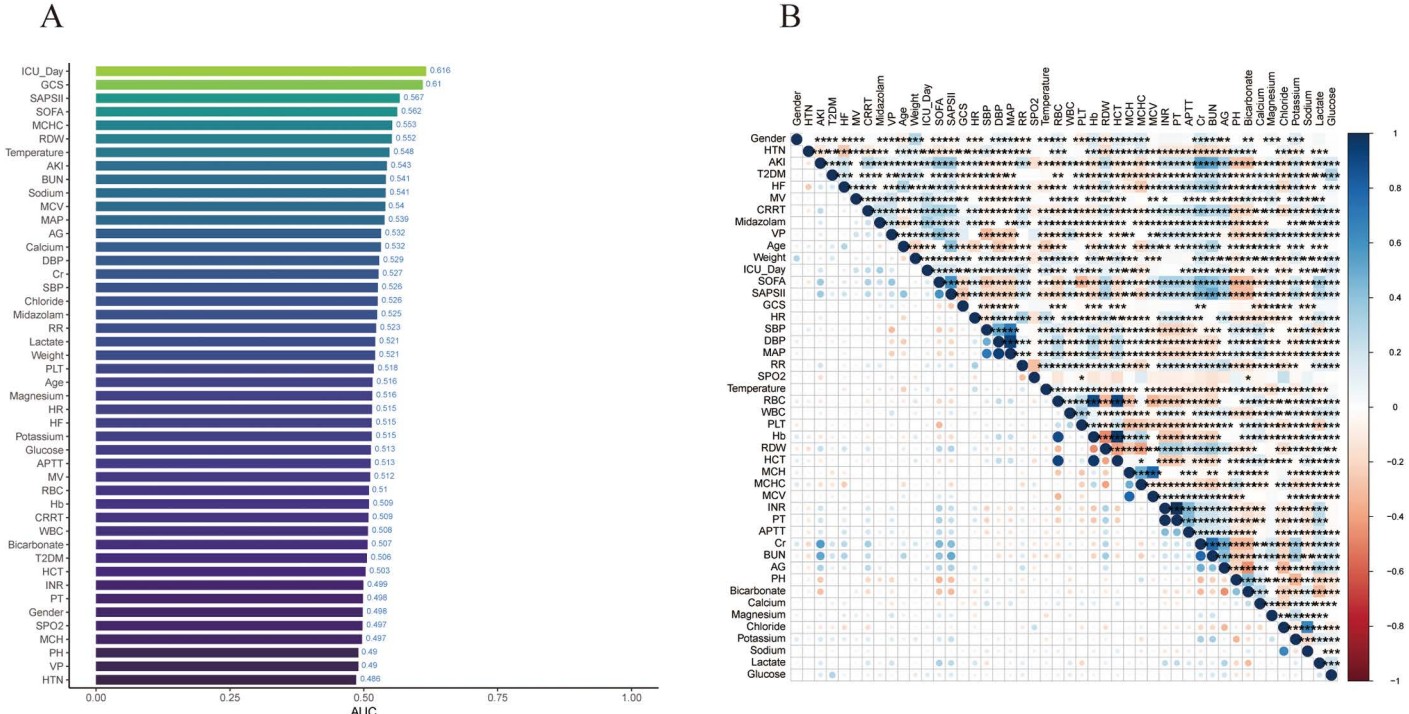

**Fig 2. The associations between variables.** (A) Bar chart of variable correlation. (B) Heat map depicting the correlations among variables.

each feature affects the model's predictions, with SHAP values plotted on the x-axis; higher SHAP values indicate an increased likelihood of the outcome occurring, and vice versa. The y-axis represents the magnitude of the feature values, visually depicted through a gradient from yellow to purple, where yellow indicates high feature values and purple signifies low feature values. According to the results, lower GCS scores, lower Chloride levels, and higher ICU Day, SOFA scores, Sodium levels, RDW, MCV, Age, and the administration of Midazolam are associated with higher SHAP values, indicating a greater likelihood of delirium in sepsis patients (Fig 5A). Fig 5B displays the GBM model's SHAP significance analysis, visualizing the ranking of feature importance. To enhance the interpretability of the model's decision-making process at the individual level, we conducted a systematic interpretability study on two representative cases. We plotted bar charts for one SAD patient and one Non-SAD patient, where yellow represents increased risk and purple indicates reduced risk (Fig 5C and 5D). Additionally, to investigate the interaction effects among variables, we generated SHAP dependence plots. Using the GCS score as an example, the SHAP values for SOFA score, serum sodium levels, and ICU day exhibited significant variation across different GCS scores. The results revealed that patients with GCS scores in the two extreme ranges (1–3 and 14–15) consistently demonstrated significantly lower delirium incidence rates compared to patients with scores in the intermediate ranges (Fig 6A–6C).

### 3.5 Construction of the online calculator

In this study, we developed an online web calculator based on the GBM model (Fig 7) (https://risk-model.shinyapps.io/make_web/). This calculator can predict the likelihood of delirium occurring in sepsis patients within the ICU based on various clinical feature variables. Clinicians can conveniently input relevant data into this tool, facilitating the use of the model to predict the incidence of SAD and allowing for timely adjustments to treatment plans to improve patient outcomes. The example parameters can be found in S2 Table.

**Table 2. Results from univariate and multivariate logistic regression analyses conducted on the training set.**

| Feature | Category | OR (univariable) | OR (multivariable) |
|---|---|---|---|
| Age | / | 1.00 (1.00–1.01, p = 0.003) | 1.01 (1.00–1.01, p = 0.013) |
| Weight | / | 1.00 (1.00–1.00, p = 0.002) | 1.00 (0.99–1.00, p < 0.001) |
| ICU day | / | 1.03 (1.03–1.04, p < 0.001) | 1.03 (1.02–1.03, p < 0.001) |
| SOFA | / | 1.05 (1.03–1.06, p < 0.001) | 1.04 (1.02–1.06, p < 0.001) |
| SAPSII | / | 1.01 (1.01–1.02, p < 0.001) | 1.00 (0.99–1.00, p = 0.764) |
| GCS | / | 0.93 (0.92–0.94, p < 0.001) | 0.94 (0.92–0.96, p < 0.001) |
| SBP | / | 1.01 (1.00–1.01, p < 0.001) | 0.99 (0.98–1.00, p = 0.005) |
| DBP | / | 1.01 (1.00–1.01, p < 0.001) | 0.98 (0.96–1.00, p = 0.025) |
| MAP | / | 1.01 (1.01–1.02, p < 0.001) | 1.04 (1.02–1.06, p < 0.001) |
| Temperature | / | 1.29 (1.18–1.41, p < 0.001) | 1.27 (1.15–1.39, p < 0.001) |
| Platelet | / | 1.00 (1.00–1.00, p < 0.001) | 1.00 (1.00–1.00, p = 0.005) |
| RDW | / | 1.07 (1.05–1.08, p < 0.001) | 1.06 (1.04–1.08, p < 0.001) |
| MCHC | / | 0.90 (0.88–0.93, p < 0.001) | 0.99 (0.96–1.03, p = 0.660) |
| MCV | / | 1.02 (1.01–1.03, p < 0.001) | 1.02 (1.01–1.02, p < 0.001) |
| Creatinine | / | 1.06 (1.03–1.09, p < 0.001) | 0.99 (0.95–1.03, p = 0.586) |
| BUN | / | 1.01 (1.00–1.01, p < 0.001) | 1.00 (1.00–1.01, p = 0.057) |
| Anion Gap | / | 1.02 (1.01–1.03, p = 0.001) | 1.00 (0.98–1.02, p = 0.917) |
| Calcium | / | 1.17 (1.10–1.24, p < 0.001) | 1.06 (0.99–1.13, p = 0.113) |
| Magnesium | / | 0.87 (0.78–0.97, p = 0.010) | 0.82 (0.72–0.93, p = 0.002) |
| Chloride | / | 0.99 (0.98–0.99, p < 0.001) | 0.96 (0.95–0.97, p < 0.001) |
| Sodium | / | 1.03 (1.02–1.04, p < 0.001) | 1.05 (1.04–1.07, p < 0.001) |
| Lactate | / | 0.94 (0.92–0.97, p < 0.001) | 0.90 (0.86–0.93, p < 0.001) |
| Glucose | / | 1.00 (1.00–1.00, p = 0.014) | 1.00 (1.00–1.00, p = 0.085) |
| AKI | No | | |
| | Yes | 1.32 (1.21–1.45, p < 0.001) | 1.14 (1.02–1.28, p = 0.018) |
| T2DM | No | | |
| | Yes | 1.11 (1.01–1.23, p = 0.037) | 1.03 (0.92–1.16, p = 0.583) |
| HF | No | | |
| | Yes | 1.12 (1.01–1.23, p = 0.028) | 0.89 (0.80–1.00, p = 0.051) |
| MV | No | | |
| | Yes | 1.21 (1.05–1.39, p = 0.009) | 1.13 (0.97–1.31, p = 0.113) |
| CRRT | No | | |
| | Yes | 1.20 (1.03–1.39, p = 0.022) | 0.82 (0.68–0.99, p = 0.041) |
| Midazolam | No | | |
| | Yes | 1.33 (1.20–1.47, p < 0.001) | 1.13 (1.00–1.27, p = 0.042) |
| VP | No | | |
| | Yes | 0.90 (0.82–0.99, p = 0.034) | 0.87 (0.77–0.98, p = 0.021) |
| Gender | Female | | |
| | Male | 1.00 (0.91–1.10, p = 0.979) | |
| Hypertension | No | | |
| | Yes | 0.91 (0.83–1.00, p = 0.058) | |
| HR | / | 1.00 (1.00–1.01, p = 0.080) | |
| RR | / | 1.01 (1.00–1.02, p = 0.137) | |
| SPO2 | / | 1.00 (0.99–1.01, p = 0.977) | |
| RBC | / | 0.94 (0.88–1.00, p = 0.056) | |
| WBC | / | 1.00 (1.00–1.01, p = 0.559) | |

*(Continued)*

**Table 2.** (Continued)

| Feature | Category | OR (univariable) | OR (multivariable) |
|---|---|---|---|
| Hemoglobin | / | 0.98 (0.96–1.00, p = 0.070) | |
| HCT | / | 1.00 (0.99–1.01, p = 0.929) | |
| MCH | / | 1.01 (0.99–1.03, p = 0.293) | |
| INR | / | 1.04 (0.99–1.10, p = 0.122) | |
| PT | / | 1.00 (1.00–1.01, p = 0.209) | |
| APTT | / | 1.00 (1.00–1.00, p = 0.124) | |
| PH | / | 0.76 (0.41–1.43, p = 0.394) | |
| Bicarbonate | / | 1.01 (1.00–1.02, p = 0.056) | |
| Potassium | / | 0.95 (0.88–1.02, p = 0.153) | |

Abbreviations: SOFA, Sequential Organ Failure Assessment; SAPSII, Simplified Acute Physiologic Score II; GCS, Glasgow Coma Scale; SBP, Systolic blood pressure; DBP, Diastolic blood pressure; MAP, Mean arterial pressure; RDW, Red cell distribution Width; MCHC, Mean corpuscular hemoglobin concentration; MCV, Mean corpuscular volume; BUN, Blood urea nitrogen; AKI, Acute kidney injury; T2DM, Type 2 diabetes mellitus; HF, Heart failure; MV, Mechanical ventilation; RBC, Red blood cell; WBC, White blood cell; HCT, Hematocrit; MCH, Mean corpuscular hemoglobin; INR, International normalized ratio; PT, Prothrombin time; APTT, Activated partial thromboplastin time.

## 4. Discussion

SAD is a severe neurological syndrome that significantly increases mortality among affected patients, leading to long-term mental and cognitive impairments, and even dementia, placing a substantial burden on patients and their families [21,22]. We developed a machine learning prediction model based on GBM, which showed promising results in internal validation (AUC: 0.73). Utilizing the SHAP method, we enhanced the clinical interpretability of the model. Although several predictive models have been developed to assess the risk of SAD in the ICU, their practicality in clinical practice remains insufficient, failing to effectively translate into clinical tools [6,7]. Therefore, we created a simple web-based calculator to assist health-care professionals in quickly identifying SAD and timely adjusting medical strategies to improve patient outcomes.

Currently, clinical recognition of SAD in the ICU is not optimistic, mainly due to physicians' insufficient understanding of the complication. This underscores the importance of enhancing clinical awareness. Through systematic training, we can improve healthcare teams' ability to recognize sepsis-associated delirium, thereby allowing for more accurate identification of early intervention opportunities and a scientific assessment of the risk-benefit ratio of treatment strategies. Commonly used delirium assessment methods in the ICU include CAM-ICU and the Intensive Care Delirium Screening Checklist (ICDSC). CAM-ICU is applicable for patients requiring mechanical ventilation, offering rapid and effective assessment; however, it is not suitable for deeply sedated or comatose patients and requires multiple evaluations for diagnosis [23–25]. In contrast, the ICDSC has a broader applicability, as it does not require patient cooperation and is based on nurses observing patients' behavior over a 24-hour period. Nonetheless, it relies on nurses' subjective judgment, which may lead to lower consistency among different evaluators and increase the nursing workload [26–28]. The time-consuming nature of both methods may delay treatment decisions and compromise patient safety. To address this issue, we constructed a web-based calculator using the GBM model to enable clinicians to quickly detect SAD early. It is important to empha-size that, although preliminary validation shows promising performance, multi-center, prospective cohort studies are still required to ensure the model's applicability in various clinical scenarios, providing external validation with independent datasets. This will help objectively assess the model's generalizability and diagnostic accuracy, offering evidence-based medicine to support its clinical translation.

Feature selection is a crucial step in building predictive models, as its validity directly affects the model's predictive performance [29]. We obtained a substantial sample from the MIMIC-IV database and utilized MLR to identify independent risk factors for SAD patients. LASSO regression was applied to process features, avoiding multicollinearity and reducing

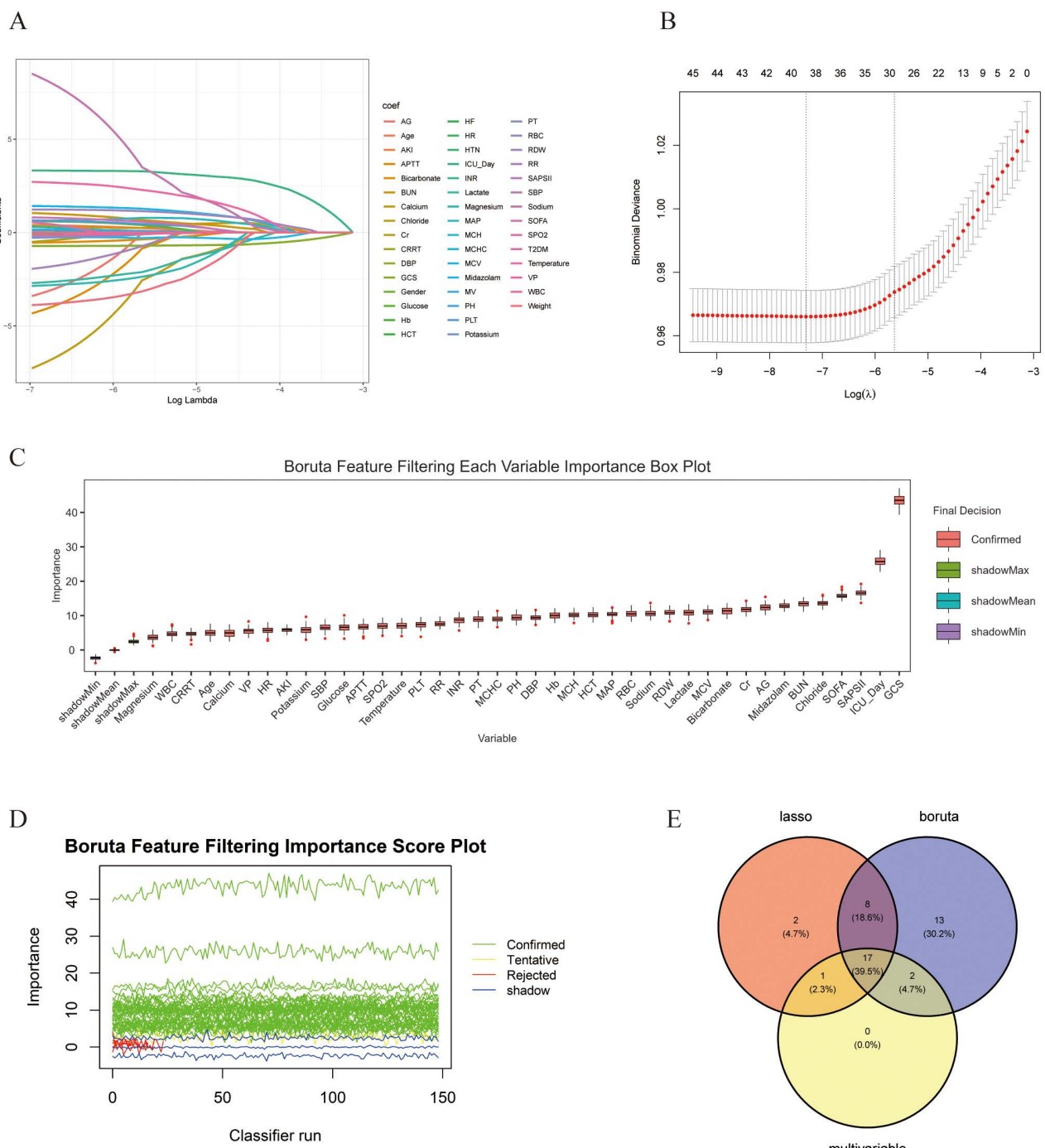

**Fig 3. Features identified through LASSO analyses and Boruta.** (A) Variable Trajectory Screening via LASSO Regression. (B) The LASSO model was subjected to 20-fold cross-validation to determine the optimal penalization parameter (lambda.1se). (C, D) Variables selected by the Boruta algorithm. In terms of feature importance scores, the 40 red variables are considered to be important variables. (E) The Venn diagram illustrates the features selected by Boruta, LASSO, and MLR. The intersection of the features identified by these three methods reveals 17 clinical characteristics.

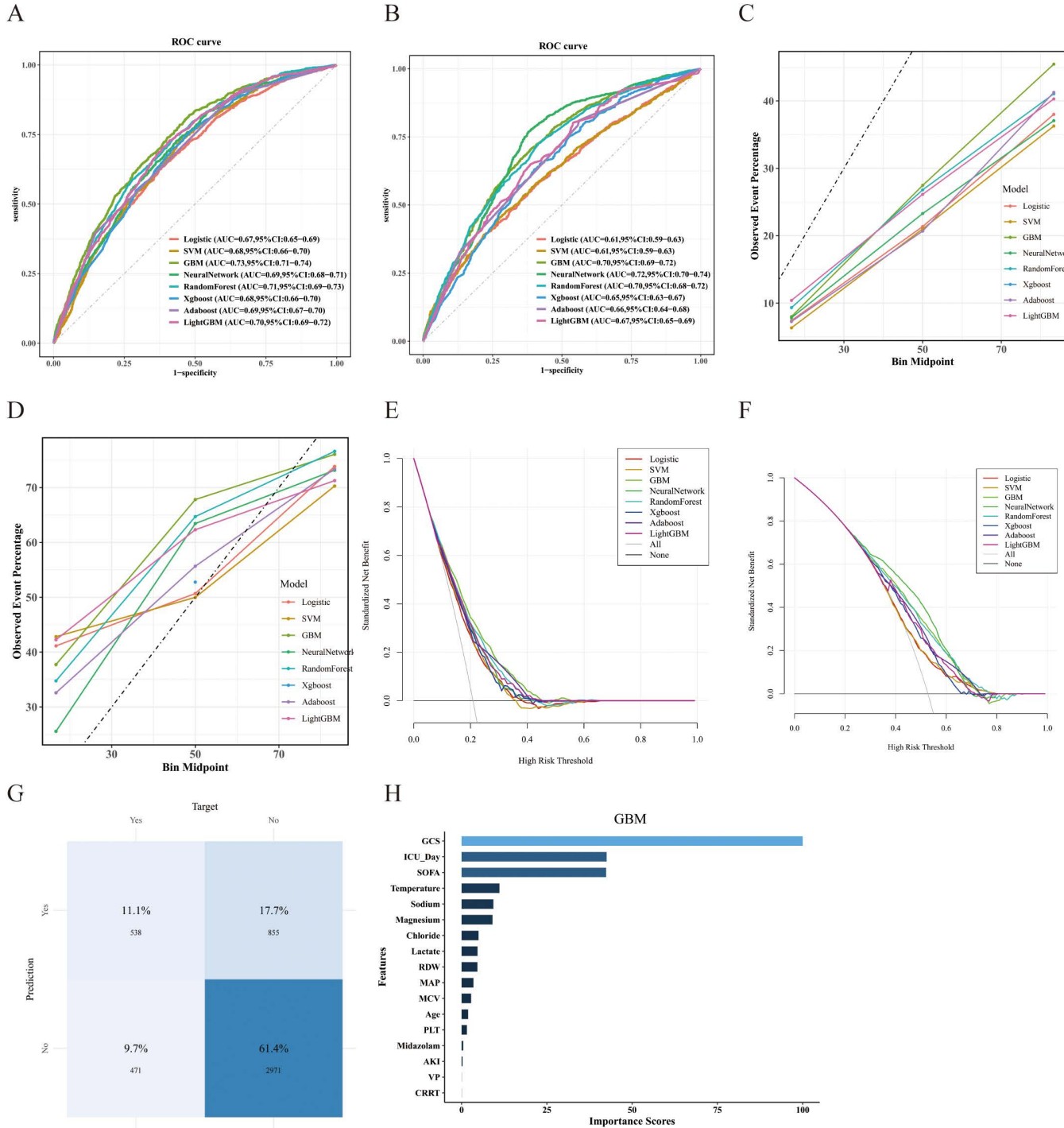

**Fig 4. Comparative analysis of the performance of eight distinct predictive models.** (A) ROC curves for the internal validation set. (B) ROC curves for the external validation set. (C, D) Calibration curves for the internal and external validation sets, respectively. (E, F) DCA curves for the internal and external validation sets, respectively. (G) Confusion matrix for the internal validation set. (H) Feature importance plot for the GBM model in the internal validation set.

**Table 3. The predictive capabilities of each model.**

| Model | Accuracy | Sensitivity | Specificity | Precision | F1-Score |
|---|---|---|---|---|---|
| Internal validation sets | | | | | |
| Logistic | 0.536 | 0.747 | 0.48 | 0.275 | 0.402 |
| SVM | 0.586 | 0.73 | 0.548 | 0.299 | 0.424 |
| GBM | 0.726 | 0.533 | 0.777 | 0.386 | 0.448 |
| NeuralNetwork | 0.606 | 0.703 | 0.581 | 0.307 | 0.427 |
| RandomForest | 0.714 | 0.509 | 0.768 | 0.367 | 0.426 |
| Xgboost | 0.675 | 0.549 | 0.708 | 0.331 | 0.413 |
| Adaboost | 0.491 | 0.862 | 0.393 | 0.273 | 0.414 |
| LightGBM | 0.68 | 0.558 | 0.712 | 0.338 | 0.421 |
| External validation sets | | | | | |
| Logistic | 0.572 | 0.692 | 0.438 | 0.579 | 0.63 |
| SVM | 0.578 | 0.646 | 0.501 | 0.591 | 0.618 |
| GBM | 0.578 | 0.318 | 0.869 | 0.731 | 0.443 |
| NeuralNetwork | 0.664 | 0.645 | 0.684 | 0.695 | 0.669 |
| RandomForest | 0.586 | 0.342 | 0.859 | 0.73 | 0.466 |
| Xgboost | 0.554 | 0.308 | 0.828 | 0.667 | 0.421 |
| Adaboost | 0.627 | 0.791 | 0.443 | 0.614 | 0.691 |
| LightGBM | 0.586 | 0.381 | 0.814 | 0.696 | 0.493 |

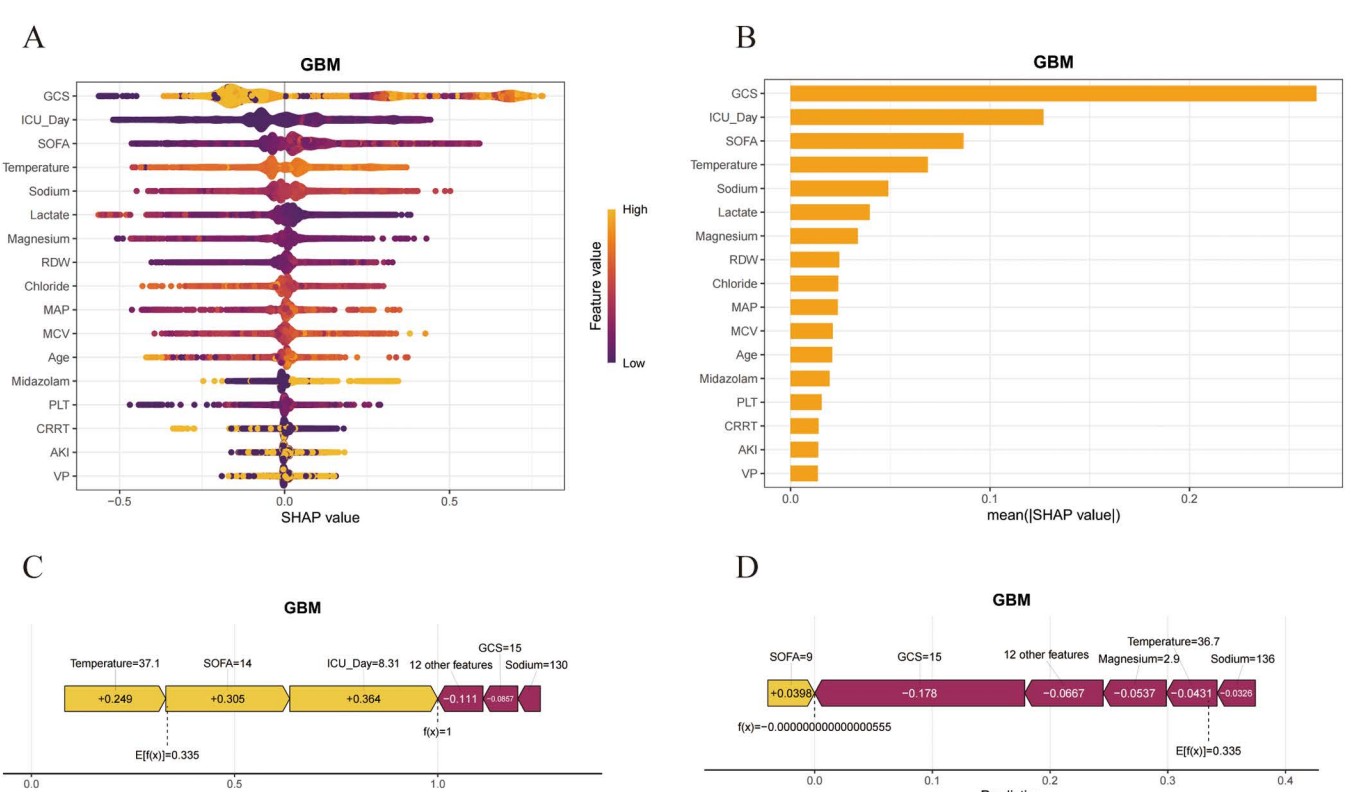

**Fig 5. SHAP analysis of the GBM model.** (A) The SHAP summary plot of the GBM model. (B) Significance analysis of feature importance ranking via SHAP, based on the mean value. (C, D) The force plots offer individualized feature attributions for two representative examples. C: Patients with SAD; D: Patients with Non-SAD.

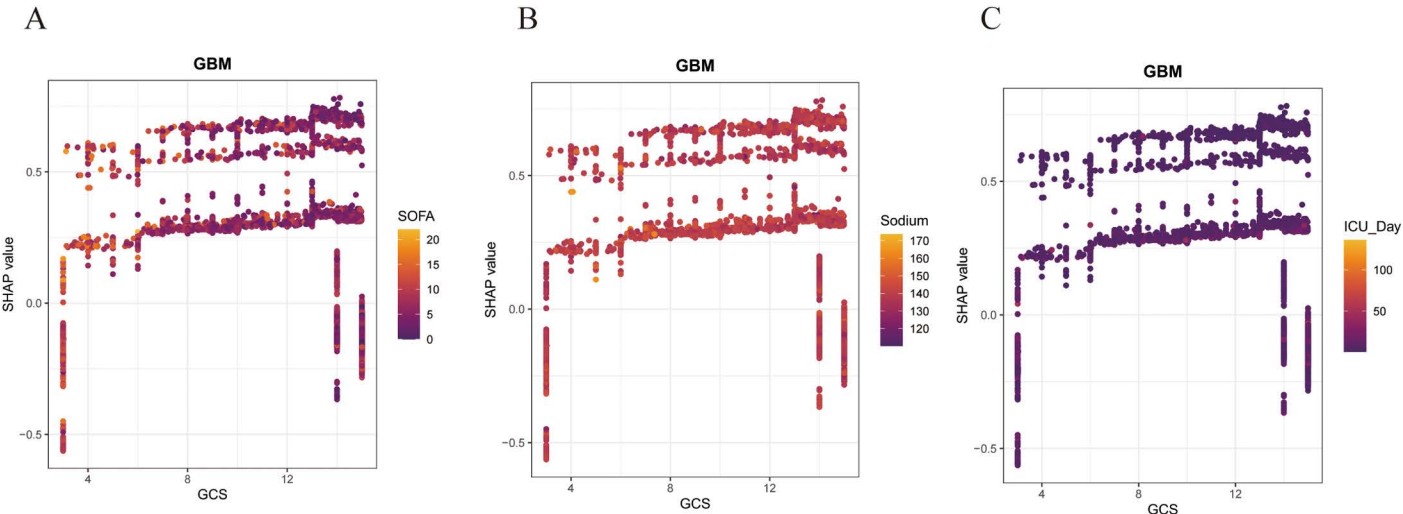

**Fig 6. The SHAP dependence plot for features in the GBM model.** (A-C) The Y-axis represents SHAP values, while the X-axis depicts actual clinical parameters. It is noteworthy that when the SHAP value of a feature is greater than 0, it indicates an increased risk of SAD, while a negative SHAP value suggests a reduced risk.

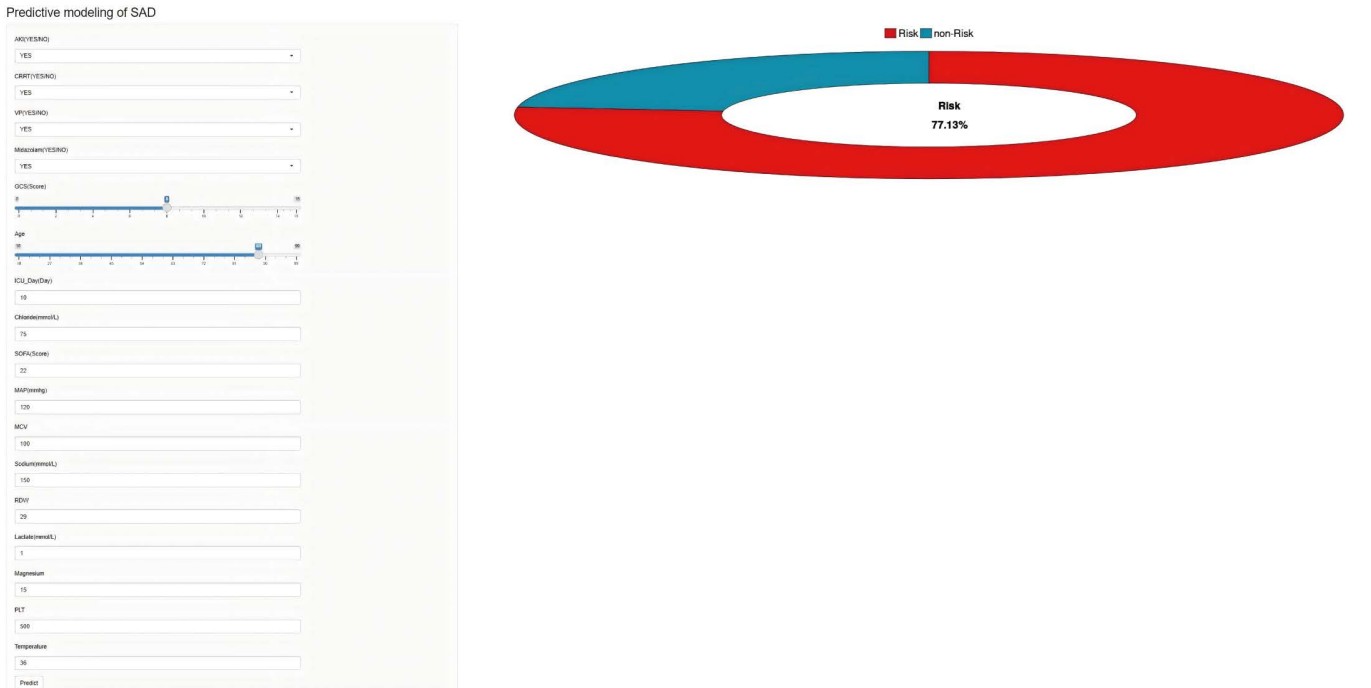

**Fig 7. Online web calculator for predicting the probability of SAD occurrence.**

the risk of overfitting [30]. Building on this approach, we employed the Boruta algorithm for feature selection, leveraging its rigorous statistical significance-based filtering mechanism to retain critical features. This methodology effectively curbs model complexity while enhancing generalization capability on independent datasets. The cross-verified features identified

through these three complementary methods deliver optimal predictive power and generalizability. This strategy ensures model robustness and parsimony, simultaneously providing an operationally efficient, computationally feasible, and clinically actionable feature set for online calculator implementation. In the realm of healthcare big data, machine learning algorithms demonstrate significant advantages in handling complex medical data [31]. Compared to traditional statistical models, the nonlinear modeling capabilities of machine learning can capture higher-order interactions between variables and enable multivariable synchronous analysis of high-dimensional data through parallel computing frameworks, thereby increasing the ability to predict diseases [32]. In this study, we compared the performance of eight machine learning models. Although RF theoretically mitigate overfitting risks through bagging and random feature selection, they can still exhibit significant overfitting in practical applications, particularly with noisy datasets or limited sample sizes. To address this, we implemented rigorous optimization procedures, including grid search, 10-fold cross-validation, and the SMOTE algorithm to alleviate sample insufficiency. Nevertheless, even after applying SMOTE to augment the training dataset, the RF model continued to demonstrate a tendency toward overfitting. Consequently, based on comprehensive performance evaluation, we ultimately identified GBM as the optimal model for this study.

However, in existing research, Zhang Y et al. utilized LASSO regression to identify key predictors for SAD patients and subsequently constructed an XGBoost prediction model based on these factors. Their study ultimately identified lower GCS scores, sedative medication use, and concomitant AKI as risk factors for SAD development, findings consistent with the results of the present study. This model demonstrated robust predictive performance on both the internal validation set (AUC = 0.793) and the external validation set (AUC = 0.701). By contrast, the AUC value of the model developed in the current study is comparatively lower. This discrepancy may stem from differences in data preprocessing pipelines and the specific feature sets incorporated. Notably, Zhang et al.'s model did not include core variables such as lactate levels and ICU_Day. The omission of such critical information may have limited its predictive capability [33]. On the other hand, Gu Q et al. employed MLR to identify higher SOFA scores, elevated lactate levels, elevated phosphate levels, and MV use as independent risk factors for SAD. They subsequently developed a nomogram for risk prediction based on these factors. However, unlike the present study, their model lacks an external validation cohort. Consequently, its generalizability remains unassessed, posing challenges for real-world clinical application and implementation [34].

GBM is one of the most representative methods in ensemble learning, which iteratively combines weak learners and utilizes gradient descent to optimize prediction errors, simulating the progressive learning and collaborative optimization characteristics found in biological systems [35–37]. As an efficient predictive model, GBM is able to capture nonlinear relationships and complex interaction effects in the data and is widely used in the construction of clinical prediction models. The clinical interpretability of machine learning models is critical for medical practice; however, interpretability has long been one of the core challenges in this field [38]. To address this issue, we adopted the SHAP method to analyze features and enhance model interpretability [39]. Compared to traditional weight-based explanatory methods, SHAP exhibits superior consistency and performance, while demonstrating greater stability across various models [40]. This study utilized SHAP value analysis, which significantly improved the model's interpretability compared to the coefficient interpretations of traditional generalized linear regression models. SHAP values not only quantify the contributions of each feature to the predictive outcomes but also provide intuitive visualizations through feature importance plots [41]. This analytical approach offers a new perspective for understanding the decision-making mechanisms of machine learning models, clearly illustrating the specific impacts of feature variables on model predictions, thereby effectively enhancing the model's interpretability and transparency.

According to the SHAP feature importance plot, a low GCS score is identified as the most significant risk factor for SAD. Studies have shown that low GCS scores, high SOFA scores, advanced age, prolonged ICU stay, hypernatremia, and the use of midazolam are all risk factors for SAD. Specifically, the GCS score is one of the strong predictive features, originally designed by Graham Teasdale and Bryan Jennett at the University of Glasgow for assessing traumatic brain injury. It quantifies eye-opening, verbal, and motor responses to objectively evaluate the degree of consciousness impairment [42].

The lower the GCS score, the higher the risk of delirium; research indicates that for each point decrease in GCS, the risk of delirium increases by approximately 34% [43,44]. Notably, the SHAP dependence plot further reveals a complex non-linear association between the GCS score and SAD. Clinically, GCS scores in the 1–3 range typically indicate severe brain injury, while scores of 14–15 signify a state of clear consciousness. Interestingly, the incidence of delirium in both these extreme scoring groups is significantly lower than in patients with intermediate scores. This phenomenon is consistent with findings from multiple prior delirium prediction models and has been reported in the literature [33]. The SOFA score is an effective tool for assessing organ dysfunction in sepsis patients, with the central nervous system score relying on the GCS. By treating GCS as an independent variable, we avoided the limitation of the SOFA score, which might focus solely on single-organ function assessment, thus more accurately reflecting brain function impairment. An elevated SOFA score signifies systemic inflammatory response, tissue hypoxia, and organ dysfunction, with central nervous system dysfunction being the most common. These factors are interrelated via a systemic inflammation-brain injury axis, where many inflammatory factors can disrupt the blood-brain barrier and subsequently induce delirium [45–47]. Research has found that patients with a SOFA score exceeding 9 have a probability of SAD occurrence greater than 70%, making the SOFA score a core indicator for predicting SAD; a high SOFA score is an independent risk factor for delirium [48].

Advanced age and prolonged ICU stay are both significant risk factors for delirium. As age increases, older patients often have various chronic diseases, malnutrition, sensory impairments, and cognitive deficits, resulting in decreased brain physiological reserve, thereby heightening the probability of delirium [49]. Patients who are in the ICU for extended periods are more likely to be exposed to mechanical ventilation, sedative medications, sleep deprivation, and infections, which can further trigger delirium. Delirious patients commonly exhibit complications such as agitation, respiratory dysfunction, and infections, which may significantly prolong their ICU stays, creating a vicious cycle. Studies have shown that remaining in the ICU for more than 7 days can transition delirium from an "acute" to a "persistent" state, with about 30%−40% of patients experiencing long-term cognitive sequelae [50,51]. Additionally, midazolam, as a benzodiazepine, suppresses the central nervous system, potentially interfering with cholinergic neurotransmission and disrupting sleep architecture, thus increasing the risk of delirium. Our findings indicate that the pre-illness use of midazolam significantly raises the risk of delirium, consistent with recent studies [52,53]. Hypernatremia also contributes to an increased incidence of delirium. Hypernatremia affects sodium-potassium pump function, leading to abnormal neural excitability and interfering with the release of inhibitory neurotransmitters such as gamma-aminobutyric acid. Furthermore, hypernatremia decreases brain energy metabolism, affecting glucose utilization and ATP production, resulting in insufficient neuronal energy, which further contributes to the onset of delirium [54–56].

In summary, the SHAP method provides significant support for personalized diagnosis and treatment of SAD patients. By quantifying the contributions of various feature variables to predictive outcomes, it enhances the interpretability of the model's predictions and offers intuitive decision-making support for clinicians. The SHAP values clearly illustrate the impact of clinical features on patient prognosis, aiding physicians in identifying key risk factors and formulating personalized intervention strategies. By integrating GBM and SHAP, this model not only improves predictive accuracy but also significantly enhances interpretability. The visualization of SHAP values allows clinicians to gain deeper insights into important features and their interactions, thereby increasing the transparency and credibility of medical decisions and providing a scientific basis for the personalized risk management of SAD. This interpretable machine learning approach holds substantial practical value in clinical settings and advances the development of precision medicine.

This study has some limitations. First, as a retrospective analysis, the nature of the study restricts the establishment of causal relationships between features and outcomes. Additionally, this study did not include key inflammatory biomarkers such as procalcitonin (PCT) and interleukin-6 (IL-6), which are closely associated with the pathophysiology of sepsis and delirium. However, due to their limited availability and high rates of missingness in the MIMIC-IV database, these variables could not be incorporated into the current model. This omission may have constrained the model's ability to capture certain inflammatory dimensions of SAD. In future work, we plan to conduct prospective, multi-center studies that

systematically collect longitudinal data on biomarkers such as PCT and IL-6. Integrating these markers may improve the predictive performance and biological interpretability of the model, thereby enhancing its clinical utility and generalizability. Lastly, the limitations of sample size reduce the performance of the machine learning algorithms. Future research should establish a systematic data collection mechanism to expand the sample size and diversity, thereby improving the predictive accuracy and reliability of the model and ensuring that the research findings have clinical application value.

## 5. Conclusion

In this study, we developed and validated an interpretable machine learning model based on the GBM algorithm to predict SAD using clinical data from the MIMIC-IV database. SHAP analysis enhanced the model's interpretability by identifying key predictors such as GCS, ICU stay duration, chloride, sodium, and SOFA score. A web-based risk calculator was also constructed to facilitate bedside risk assessment. The model demonstrates promising predictive accuracy and practical utility, offering a valuable tool for early identification of SAD and supporting individualized clinical decision-making.

## Supporting information

**S1 Table. Hyperparameter settings for eight models.** Gradient Boosting Machine: GBM; Support Vector Machine: SVM; Random Forest: RF; Extreme Gradient Boosting: XGBoost; Adaptive Boosting: AdaBoost; Light Gradient Boosting Machine: LightGBM.
(DOCX)

**S2 Table. Example input parameters for the online risk calculator predicting SAD.**
(DOCX)

**S3 Table. Baseline characteristics of SAD and non-SAD patients in the external validation cohort.**
(DOCX)

## Acknowledgments

The authors sincerely thank the MIMIC official team for their outstanding contributions, which are of significant importance.

## Author contributions

**Conceptualization:** Lang Gao.

**Data curation:** Lang Gao, Xu Jie Wang.

**Methodology:** Xu Jie Wang, Yun Ruo Chen.

**Project administration:** Yun Ruo Chen.

**Resources:** Lang Gao, Xing Yi Yang, Jin Ying Bai.

**Software:** Lang Gao.

**Supervision:** Xing Yi Yang, Shi Jun Tong, Jin Ying Bai.

**Writing – original draft:** Lang Gao, Guang Dong Wang, Shi Jun Tong, Ya Xin Zhang.

**Writing – review & editing:** Lang Gao, Guang Dong Wang, Ya Xin Zhang.

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
