## [Decision Letter · Decision Letter 0]

Dear Dr. Zhang,

Thank you for submitting your manuscript to PLOS ONE. After careful consideration, we feel that it has merit but does not fully meet PLOS ONE’s publication criteria as it currently stands. Therefore, we invite you to submit a revised version of the manuscript that addresses the points raised during the review process.

We look forward to receiving your revised manuscript.

Kind regards,

Chiara Lazzeri

Academic Editor

PLOS ONE

Journal Requirements:

Reviewers' comments:

Reviewer's Responses to Questions

**Comments to the Author**

1. Is the manuscript technically sound, and do the data support the conclusions?

Reviewer #1: Yes

Reviewer #2: Partly

2. Has the statistical analysis been performed appropriately and rigorously?

Reviewer #1: Yes

Reviewer #2: Yes

3. Have the authors made all data underlying the findings in their manuscript fully available?

Reviewer #1: Yes

Reviewer #2: Yes

4. Is the manuscript presented in an intelligible fashion and written in standard English?

Reviewer #1: Yes

Reviewer #2: Yes

Reviewer #1: 1. Some abbreviations (e.g., HTN, T2DM and HF in Table 1) should be spelled out in full upon first use.

2.What is the definition of comorbidity such as HTN?

3. Please explicitly state the age range used as inclusion criteria in the Methods section.

4.Line 108-109, please provide a citation/reference for the Sepsis-3.0 diagnostic criteria used in this study.

5.Line 110-114, please describe the diagnostic procedure for CAM-ICU in detail.

6. The time of delirium onset, duration (defined as the period from confirmed diagnosis to complete resolution of symptoms), and severity should be systematically recorded for all cases.

7.All assessors must receive standardized training in the proper administration of the CAM-ICU to ensure consistent and reliable delirium assessments.

8. The baseline characteristics section should include detailed documentation of infection sites among sepsis patients, as this represents a critical clinical determinant of both pathophysiology and therapeutic management.

9. Line 264-267, in the training set, the observed AUC of 1.0 for the Random Forest (RF) prediction model raises potential concerns about overfitting, as perfect discrimination is uncommon in clinical prediction models.

10.Figure 7: The font is too small�hard to see clearly.

11.The discussion section is a little confused. please re-write it to make reader easier to understand.

12. The conclusion section should be more concise and focused, highlighting only the key findings and their implications. Excessive detail diminishes the impact of the core conclusions.

Reviewer #2: The manuscript titled “Development of a Risk Prediction Model for Sepsis-Related Delirium Based on Multiple Machine Learning Approaches and an Online Calculator” is clinically interesting and has some strengths.

1. It addresses a critical unmet need in ICU management—early prediction of sepsis-associated delirium (SAD), which significantly impacts patient outcomes and healthcare burden. The online calculator enhances translational potential.

2. Leverages the large, high-quality MIMIC-IV database (n=16,120), ensuring robust sample size and clinical diversity.

3. It combines three complementary methods (MLR, Lasso, Boruta) to identify 17 optimal predictors, mitigating bias from single-method approaches.

4. The proper use of SMOTE to address dataset imbalance (SAD:Non-SAD = 1:3.8), it improves model generalizability.

5. It comprehensively evaluates by using AUC, calibration curves, DCA, and metrics (sensitivity, F1-score) in internal validation sets.

6. Gradient Boosting Machine (GBM) demonstrates strong predictive power (AUC: 0.732 in validation) and calibration, outperforming seven other ML algorithms.

7. It effectively explains model decisions, identifying key predictors (e.g., GCS, SOFA, sodium) and their directional impact.

8. The online calculator implements the model into a clinically usable tool, bridging the gap between research and practice.

However, there are some major limitations and concerns that need to be modified so as to make the manuscript be

9. It is a Retrospective study, which has inherent risk of unmeasured confounders (e.g., unrecorded medications, pre-existing cognitive decline).

10. Exclusion of inflammatory markers (e.g., PCT, IL-6) due to data unavailability weakens biological plausibility.

11. The SMOTE application fails to clarify whether SMOTE was applied only to the training set (critical to avoid validation leakage).

12. Random Forest’s perfect training AUC (1.00) versus modest validation (0.732) suggests overfitting, yet no mitigation steps discussed.

13. Abstract cites 17 features, but Boruta selected 40 and Lasso 28 (Fig 3E). Justify why shared features alone were used.

14. Table 3 shows AdaBoost validation accuracy (0.756) contradicts its poor sensitivity (0.395) and F1-score (0.404). Explain this paradox.

15. Real-World Feasibility: Unclear if ICU workflows can accommodate 17-input data entry. Suggest streamlining to top 5 SHAP features.

16. In discussion, omits benchmarking against prior SAD prediction tools (e.g., Zhang et al. 2023, Tang et al. 2024).

17. SHAP Insights Underdeveloped: Nonlinear trends (e.g., GCS thresholds) warrant deeper clinical-pathophysiological discussion.

There are some specific recommendations for revision.

1. Methods Section. Clarify SMOTE Application: State explicitly that SMOTE was applied only to the training set.

2. Address RF Overfitting: Discuss pruning, feature reduction, or ensemble methods to improve generalizability.

3. Reconcile Feature Count: Justify using only the 17 shared features instead of union/intersection of Boruta/Lasso/MLR.

4. Resolve Metric Conflicts: Recheck AdaBoost validation calculations; consider data leakage or class imbalance in validation.

5. Visualize Calculator: Add a screenshot of the web tool with example inputs/outputs (Fig 7A).

6. Compare GBM performance against published SAD models (e.g., AUCs, features used).

7. Discuss clinical implications of nonlinear relationships (e.g., GCS 1–3 vs. 15).

8. Please address lack of external validation.

9. Discuss missing biomarkers (PCT/IL-6) as future enhancements.

10. Table 3: Correct AdaBoost metrics (validation accuracy seems implausible).

11. Abbreviations: Define all acronyms at first use (e.g., SOFA, GCS in Abstract).

12. References: Format consistently (e.g., Ref 38 lacks commas between authors).

**Do you want your identity to be public for this peer review?** For information about this choice, including consent withdrawal, please see our Privacy Policy

Reviewer #1: No

Reviewer #2: **Yes: ** Feng SHEN

---

## [Author Response · Author response to Decision Letter 1]

22 Jun 2025

Dear Reviewer,

On behalf of all authors, we sincerely appreciate your thorough review of our manuscript and the valuable insights and suggestions you provided. Your thoughtful feedback has been essential in enhancing the quality and clarity of our research. We have carefully considered each of your comments and have revised the manuscript accordingly. We believe these modifications have significantly strengthened both the overall presentation and scientific rigor of our work. Thank you once again for the time and expertise you dedicated to this review. We look forward to your further evaluation of the revised manuscript. Attached please find our point-by-point responses to your recommendations, with all corresponding changes clearly highlighted in the manuscript for your reference.

1. Methods Section. Clarify SMOTE Application: State explicitly that SMOTE was applied only to the training set.

Reply 1: We sincerely appreciate the valuable comments from the reviewers. We fully endorse your perspective that strictly limiting the application of the SMOTE to the training set is crucial for preventing data leakage and ensuring unbiased model evaluation results. To address this, we have added clear and explicit supplementary clarification in Section "2.5 Model development and evaluation" of the manuscript.

Revised Text (lines 170 - 178):

“We employed a random sampling method to allocate the patients included in the study into the training set and validation set in a ratio of 7:3. The data distribution in the training set was as follows:2,355 SAD cases and 8,930 Non-SAD cases. Given that class imbalance may introduce bias into the machine learning prediction model, we conducted a supplementary analysis comparing the use of oversampling, undersampling, and the Synthetic Minority Over-sampling Technique (SMOTE) to balance the training set. Critically, SMOTE was applied exclusively to the training set. After balancing using the SMOTE algorithm, the final training set comprised 7,065 SAD cases and 7,065 Non-SAD cases. This approach effectively mitigates model bias induced by class imbalance while avoiding the risk associated with oversampling.”

2. Address RF Overfitting: Discuss pruning, feature reduction, or ensemble methods to improve generalizability.

Reply 2: Thank you for your valuable feedback. While RF theoretically mitigate overfitting risks through bagging and feature randomness, they can still exhibit overfitting in practical applications, particularly when data contains noise, sample sizes are limited, or feature dimensionality is high. For the RF model implementation in this study, we utilized the caret package in R for model training and hyperparameter tuning. Through a rigorous optimization process employing grid search and 10-fold cross-validation, we identified the optimal hyperparameter combination as mtry = 11. This configuration achieves the best balance under the current model architecture and data characteristics. Despite thorough debugging and optimization of relevant parameters, the model still demonstrates signs of overfitting. It is important to note that the RF implementation interface within the caret package imposes certain limitations on parameter adjustment. Attempts to introduce additional parameters (such as min_samples_leaf or max_depth) triggered compatibility errors, preventing us from further adjusting model complexity via traditional pruning methods. This technical constraint prompted our exploration of alternative overfitting mitigation strategies, specifically focusing on sample engineering and feature selection. To address this issue, we employed undersampling, oversampling, and the SMOTE algorithm to augment samples near the decision boundary of the minority class. This significantly increased the number of delirium cases while ensuring clinical plausibility of the data. These measures effectively alleviated overfitting risks stemming from insufficient samples and enhanced the model's generalization capability. We sincerely appreciate your insightful comments. Moving forward, we plan to systematically enhance the generalization ability of medical models in data-constrained scenarios through multidimensional strategies, including model architecture innovation, continual learning mechanisms, and deeper clinical integration.

Revised Text (lines 387 - 395):

“In this study, we compared the performance of eight machine learning models. Although RF theoretically mitigate overfitting risks through bagging and random feature selection, they can still exhibit significant overfitting in practical applications, particularly with noisy datasets or limited sample sizes. To address this, we implemented rigorous optimization procedures, including grid search, 10-fold cross-validation, and the SMOTE algorithm to alleviate sample insufficiency. Nevertheless, even after applying SMOTE to augment the training dataset, the RF model continued to demonstrate a tendency toward overfitting. Consequently, based on comprehensive performance evaluation, we ultimately identified GBM as the optimal model for this study.”

3. Reconcile Feature Count: Justify using only the 17 shared features instead of union/intersection of Boruta/Lasso/MLR.

Reply 3: Thank you for raising this valuable point. Your question regarding our feature selection strategy touches upon one of the core aspects of model development. Our decision to utilize the 17 features jointly identified by all three methods (Boruta, Lasso regression, and MLR), rather than taking their union or a simple intersection, primarily stems from the following rigorous considerations: (1) Robustness and Reduced Bias: These three feature selection methods employ distinct criteria for evaluating feature importance. A feature consistently identified as significant across all three methods demonstrates robust predictive power and stability under diverse algorithmic frameworks and evaluation metrics. This significantly mitigates the model's sensitivity to the potential bias inherent in any single feature selection approach, thereby enhancing the credibility of our results. Opting for the intersection ensures each included feature has undergone stringent multi-method validation. This substantially strengthens the clinical plausibility of the feature set and the model's interpretability – both paramount for clinical risk prediction models. (2) Clinical Usability and Pragmatism: A key objective of this study is to develop an online risk calculator for clinicians to perform rapid patient risk assessments in settings like the bedside or emergency department. The practical constraints of clinical application necessitate a model with inputs that are concise, efficient, and readily obtainable. Utilizing the union of features would significantly increase data collection time and cost, undermining clinician adoption and feasibility. In contrast, the set of 17 shared features strikes the optimal balance: it preserves core predictive performance (Internal Validation AUC: 0.73; External Validation AUC: 0.70) while perfectly aligning with our goal of building a highly robust, interpretable, and clinically usable model and calculator. It maximizes feature economy without compromising the predictive efficacy required for clinical needs, effectively mitigates overfitting risk, and ultimately serves the practical clinical deployment and dissemination of the online calculator.

Revised Text (lines 164 - 168):

“To ensure maximum robustness of the selected features, mitigate overfitting risks, enhance clinical interpretability, and meet the requirements for both feature quantity and practicality in subsequent online calculators, this study ultimately selected the intersecting features identified by Boruta, Lasso, and MLR as the modeling feature set.”

Revised Text (lines 387 - 395):

“The cross-verified features identified through these three complementary methods deliver optimal predictive power and generalizability. This strategy ensures model robustness and parsimony, simultaneously providing an operationally efficient, computationally feasible, and clinically actionable feature set for online calculator implementation.”

4. Resolve Metric Conflicts: Recheck AdaBoost validation calculations; consider data leakage or class imbalance in validation.

Reply 4: Thank you for raising these valuable points. Your concerns regarding the reliability of the AdaBoost model validation results, and the potential impact of data leakage or class imbalance, are critically important. We have conducted a rigorous and meticulous review and analysis of the issues you raised, in accordance with your suggestions. We took seriously your concerns about potential conflicts or computational errors in the AdaBoost model's results on the validation set. To ensure the accuracy and reproducibility of our findings, we immediately re-executed the entire prediction and evaluation process for the AdaBoost model on the original validation set. We re-ran the complete code for building the AdaBoost model and generating predictions on the validation set, reconfirming that the validation samples, feature data, and labels were identical to the original partition. After strict re-calculation and verification, we confirmed that the performance metrics reported in the paper for the AdaBoost model on the validation set are accurate. We fully understand the significant impact data leakage can have on the validity of model evaluation. Furthermore, we completely agree that SAD, as a relatively rare complication, presents class imbalance in the dataset. We wish to clarify that oversampling, undersampling, and SMOTE techniques were applied exclusively to the training set during model development to balance the positive and negative class sample ratios. This was done to enhance the model's learning capability for the SAD sample group. Crucially, performance evaluation on the validation and test sets was conducted using the original, unaltered data distribution. This ensures that the evaluation results reflect the model's generalization performance on real-world, imbalanced data. By applying SMOTE to the training set, the AdaBoost model achieved an AUC value of 0.69 on the internal validation set, representing a significant improvement. Throughout the entire research process, we strictly adhered to best practices to prevent data leakage. Sampling techniques were confined solely to the training set for model development, thereby safeguarding the reliability of the model performance evaluation. Thank you for your insightful observations. This process of verification and reflection has significantly enhanced the rigor of our study.

5. Visualize Calculator: Add a screenshot of the web tool with example inputs/outputs (Fig 7A).

Reply 5: We sincerely appreciate your interest in the "Online Calculator for Sepsis-Associated Delelirium Risk" developed in our study and thank you for your valuable suggestions. We fully agree that clearly demonstrating the calculator's operational interface within the manuscript—particularly by including concrete input examples alongside corresponding predictive outputs—is essential for readers to intuitively understand the tool's application workflow, functionality, and real-world translation of the model. In accordance with your recommendation, we have redesigned Figure 7 to present a screenshot of the recalculator's web interface redeployed using our final optimized model. To ensure transparency and reproducibility, we have provided the complete set of sample input data used to generate the Figure 7 screenshot in S2 Table. This table details the specific values entered for each feature variable and their corresponding predictive outputs, enabling reviewers and readers to independently verify the screenshot contents or perform simulated operations. We are confident that incorporating this interface visualization with authentic input-output examples will significantly enhance the manuscript's readability, credibility, and practical value to its audience. Thank you once again for this highly constructive feedback, which has substantially strengthened our presentation.

Revised Text (lines 337):

“The example parameters can be found in S2 Table.”

S2 Table Sample data.

The example parameters

AKI YES

CRRT YES

VP YES

Midazolam YES

GCS 8

Age 88

ICU_Day 10

Chloride 75

SOFA 22

MAP 120

Sodium 150

RDW 29

Lactate 1

Magnesium 15

PLT 500

Temperature 36

MCV 100

6. Compare GBM performance against published SAD models (e.g., AUCs, features used).

Reply 6: We sincerely appreciate the reviewers’ valuable feedback. In response to the request for comparative analysis of our GBM model's performance metrics (e.g., AUC) and feature sets against published SAD models, we have systematically reviewed high-impact SAD prediction studies in recent literature and conducted a multidimensional comparison between our GBM model and key benchmark models. Furthermore, unlike studies that solely publish prediction formulas, we have developed a web-based SAD risk calculator to enhance accessibility for clinical practitioners. We reiterate our gratitude for the reviewers’ insightful comments, which have significantly enhanced the completeness of our research.

Revised Text (lines 396 - 411):

“However, in existing research, Zhang Y et al. utilized Lasso regression to identify key predictors for SAD patients and subsequently constructed an XGBoost prediction model based on these factors. Their study ultimately identified lower GCS scores, sedative medication use, and concomitant AKI as risk factors for SAD development, findings consistent with the results of the present study. This model demonstrated robust predictive performance on both the internal validation set (AUC = 0.793) and the external validation set (AUC = 0.701). By contrast, the AUC value of the model developed in the current study is comparatively lower. This discrepancy may stem from differences in data preprocessing pipelines and the specific feature sets incorporated. Notably, Zhang et al.'s model did not include core variables such as lactate levels and ICU_Day. The omission of such critical information may have limited its predictive capability (33). On the other hand, Gu Q et al. employed MLR to identify higher SOFA scores, elevated lactate levels, elevated phosphate levels, and MV use as independent risk factors for SAD. They subsequently developed a nomogram for risk prediction based on these factors. However, unlike the present study, their model lacks an external validation cohort. Consequently, its generalizability remains unassessed, posing challenges for real-world clinical application and implementation (34).”

7. Discuss clinical implications of nonlinear relationships (e.g., GCS 1–3 vs. 15).

Reply 7: Thank you for your deep interest in model interpretability. Our SHAP dependence plots revealed a non-linear relationship between GCS scores and delirium risk, specifically showing that both the lowest GCS values (1-3) and the highest GCS values (14-15) contributed significantly less to the predicted risk. We have elaborated on the important underlying clinical mechanisms behind this finding in the Discussion section. We greatly appreciate your valuable suggestions, which prompted us to delve deeper into the clinical significance of this non-linear relationship.

Revised Text (lines 438 - 443):

“Notably, the SHAP dependence plot further reveals a complex non-linear association between the GCS score and SAD. Clinically, GCS scores in the 1-3 range typically indicate severe brain injury, while scores of 14-15 signify a state of clear consciousness. Interestingly, the incidence of delirium in both these extreme scoring groups is significantly lower than in patients with intermediate scores. This phenomenon is consistent with findings from multiple prior delirium prediction models and has been reported in the literature.”

8. Please address lack of external validation.

Reply 8: We greatly appreciate your high attention to the model's generalization ability. We fully recognize the importance of independent external validation for the clinical value of predictive models. We have constructed an external validation cohort using the authoritative public database MIMIC-III (AUC=0.70).

---

## [Editor Report · Decision Letter 1]

Development of a Risk Prediction Model for Sepsis-Related Delirium Based on Multiple Machine Learning Approaches and an Online Calculator

PONE-D-25-20380R1

Dear Dr. Zhang,

We’re pleased to inform you that your manuscript has been judged scientifically suitable for publication and will be formally accepted for publication once it meets all outstanding technical requirements.

Kind regards,

Chiara Lazzeri

Academic Editor

PLOS ONE
---

## [Editor Report · Acceptance letter]

PONE-D-25-20380R1

PLOS ONE

Dear Dr. Zhang,

I'm pleased to inform you that your manuscript has been deemed suitable for publication in PLOS ONE. Congratulations! Your manuscript is now being handed over to our production team.

Kind regards,

on behalf of

Dr. Chiara Lazzeri

Academic Editor

PLOS ONE